# Assessing the Impact of Deforestation on Decadal Runoff Estimates in Non-Homogeneous Catchments of Peninsula Malaysia

**Jen Feng Khor [1], Steven Lim [2], Vania Lois Ling [3] and Lloyd Ling [1],***

[1]  Centre of Disaster Risk Reduction (CDRR), Lee Kong Chian Faculty of Engineering & Science, Universiti Tunku Abdul Rahman, Jalan Sungai Long, Kajang 43000, Malaysia
[2]  Centre for Photonics and Advanced Materials Research, Lee Kong Chian Faculty of Engineering & Science, Universiti Tunku Abdul Rahman, Jalan Sungai Long, Kajang 43000, Malaysia
[3]  School of Banking and Finance, Asia Pacific University, Technology Park Malaysia, Kuala Lumpur 57000, Malaysia
*  Correspondence: linglloyd@utar.edu.my

**Abstract:** This study calibrated the Soil Conservation Service Curve Number (SCS-CN) model to predict decadal runoff in Peninsula Malaysia and found a correlation between the reduction of forest area, urbanization, and an increase in runoff volume. The conventional SCS-CN runoff model was found to commit a type II error in this study and must be pre-justified with statistics and calibrated before being adopted for any runoff prediction. Between 1970 and 2000, deforestation in Peninsula Malaysia caused a decline in forested land by 25.5%, resulting in a substantial rise in excess runoff by 10.2%. The inter-decadal mean runoff differences were more pronounced in forested and rural catchments (lower CN classes) compared to urban areas. The study also found that the CN value is a sensitive parameter, and changing it by ±10% can significantly impact the average runoff estimate by 40%. Therefore, SCS practitioners are advised not to adjust the CN value for better runoff modeling results. Additionally, NASA's Giovanni system was used to generate 20 years of monthly rainfall data from 2001–2020 for trend analysis and short-term rainfall forecasting. However, there was no significant uptrend in rainfall within the period studied, and occurrences of flood and landslide incidents were likely attributed to land-use changes in Peninsula Malaysia.

**Keywords:** deforestation and decadal runoff predictions; urbanization; non-homogenous catchments; Peninsula Malaysia

## 1. Introduction

Humans have significantly altered the earth's surface over time by transforming natural areas into industrial or agricultural regions. In the case of Malaysia, as a developing country, there has been extensive land-use and land-cover change since the 1970s. The rapid pace of development in Peninsula Malaysia has resulted in changes to the landscape and vegetation. These activities have had an uneven impact on the drainage basins, leading to changes in runoff patterns. Human activities, especially deforestation, have a greater impact on vegetation cover and the environment at local, regional, and global levels. Land-use and water cycle patterns are also influenced by anthropogenic activities, particularly deforestation, which has cleared the way for urban development. As a result, it is crucial to explore and estimate the impact of development activities on runoff patterns in rural catchments in Malaysia [1–3].

Forests are an essential element in sustaining our supply of water [4–8]. However, to make room for development, many forested areas have been cleared [9–15]. Between 2002 and 2020, Malaysia lost 27,000 km$^2$ (roughly 2/3 of Switzerland or nearly 39 Singapores) [16] of humid primary forest area at the clearing rate of 1421 km$^2$/year within

19 years. This had resulted in a 17% reduction of humid primary forest in Malaysia within the given time period [17]. Compared to the deforestation of an area of 13,000 km$^2$ from 1978 to 1994 [18], the forest-clearing rate in Malaysia has doubled in recent decades. Deforestation challenged the quality and quantity of water [19] and caused significant hydrological changes which included an increase in runoff [20,21].

Interception losses in tropical and subtropical rainforests have been shown to vary from 6 to 42% of rainfall [22–24]. Meanwhile in Malaysia and Indonesia, forest interception loss was reported from 12.7% to 21% [25,26]. Forest removal will convert the interception losses into a contributing factor to increase surface runoff. Past research studies have investigated rainforest conversion and its hydrological impact [27–34]. Zhang et al. [35] reported that a reduction of forest cover from 22% to 10% caused changes in river discharge in China. Sharma et al. [36] concluded that under projected land-use scenarios, runoff would increase when forest areas were converted into agricultural land in the central Himalayas.

The conversion of natural land has increased the surface runoff while rapid urbanization and industrialization have been cited as the main causes of major flooding in Malaysia [37,38]. Many researchers quantified the effects on hydrological parameters as due to changes in vegetation cover or forest logging activities [39–44], while urbanization was also identified to be a key factor in landscape alteration, causing an increase in runoff. River discharge can also be affected by human interventions. The landscape alteration caused faster runoff from storms, increased peak flows and changed hydrologic cycles [45–47]. Sahin and Hall [48] reported that a 10% reduction in canopy cover resulted in a 20 to 25 mm increase in annual water yield while Bosh and Heewlett [39] found that a 10% removal increased the water yield by an average of 40 mm. Although flooding is a response to the complex hydrological system, the dynamic changes are caused by anthropogenic factors to mother nature.

The Soil Conservation Services (SCS) of the United States has developed a widely accepted methodology, called the Curve Number (CN), to estimate the direct runoff from rainfall in hydrological studies. This method is based on the concept that the amount of runoff produced by a rainfall event depends on various factors, such as land use, soil type, and the antecedent moisture condition of the soil. The SCS-CN method uses a CN value that represents the combined effect of these factors on runoff generation, ranging from 0 to 100. This value is used in an equation to estimate the amount of direct runoff from rainfall. The SCS-CN method is commonly used in watershed management, flood forecasting, and erosion control planning, and has been widely adopted by government agencies worldwide, integrated into software, and taught in hydrology courses.

However, studies around the world in recent decades reported runoff prediction accuracy problems with the conventional SCS-CN model [49,50]. Previous studies have utilized the SCS Curve Number (CN$_{0.2}$) to demonstrate variations in runoff response due to agricultural land-use and seasonal changes [51]. This paper applied the SCS-CN model calibration methodology with inferential statistics that was developed by authors in a previous study [50] and demonstrated the extended application to model decadal rainfall-runoff conditions in Peninsula Malaysia. The correlation between deforestation and urbanization on runoff increment in Peninsula Malaysia was also established.

## 2. Materials and Methods

### 2.1. Study Site and Data Collection

Peninsula Malaysia, with a land area of 132,265 km$^2$, is slightly larger than England (130,395 km$^2$). It is bordered by Thailand to the north and Singapore across the strait of Johor to the south. This study utilized the calibrated SCS lump rainfall-runoff model in conjunction with the most recent rainfall-runoff dataset published by the Malaysian federal agency. The dataset, which can be located in the appendix of the Department of Irrigation and Drainage's Hydrological Procedures no. 27 (DID HP 27), documented 227 storm events across 41 distinct catchments between 1970 and January 2000 in Peninsula Malaysia [52]. The smallest recorded storm event had a rainfall depth of 19 mm, with a measurable runoff

depth of 4.8 mm, while the largest event measured 420 mm in rainfall depth and 258 mm in runoff depth.

In this study, the DID HP 27 dataset was divided into three periods to investigate the potential relationship between decadal runoff conditions, urban population growth in Peninsula Malaysia, and deforestation data provided by the Forestry Department Peninsula Malaysia [53–59]. This study grouped 58 recorded storm events from 1970 to 1979 as the M70 dataset, 81 events from 1980 to 1989 as the M80 dataset, and 88 events from 1990 to January 2000 as the M90 dataset for runoff analyses and comparison.

### 2.2. Calibration of SCS-CN Model

The SCS initially developed the CN rainfall-runoff model (Equation (1)) for the federal flood control program in 1954, and it has since become the basic model for runoff estimation.

$$Q = \frac{(P - I_a)^2}{P - I_a + S} \tag{1}$$

$Q$ = Runoff depth (mm)
$P$ = Rainfall depth (mm)
$I_a$ = Rainfall initial abstraction amount (mm)
$S$ = Catchment maximum water retention potential (mm)
where $I_a = \lambda S$. If $P < I_a$, $Q = 0$.

The SCS also put forth the hypothesis that $I_a = \lambda S = 0.2S$, where $\lambda$ represents the initial abstraction ratio coefficient, which was proposed as a constant value of 0.2. The justification for Equation (1) was based on daily rainfall and runoff data, rather than event measurements, and its only official documentation source can be found in the National Engineering Handbook, Section 4 (NEH-4) [60,61]. By substituting $I_a = 0.2S$, Equation (1) is simplified into Equation (2):

$$Q = \frac{(P - 0.2S)^2}{P + 0.8S} \tag{2}$$

DID HP 27 dataset was used to assess the SCS-CN rainfall-runoff model in authors' previous study [50] with inferential statistics and concluded that the existing SCS-CN model is not even statistically significant at alpha = 0.05 level for runoff predictions in Peninsula Malaysia. Therefore, it must be calibrated according to the local rainfall-runoff dataset and the key parameter $\lambda$ must be derived to formulate a statistically significant rainfall-runoff model for Peninsula Malaysia. All statistical analyses were performed using the IBM SPSS software version 26.0 in this study [62].

The SCS model was calibrated according to each decadal rainfall-runoff data batch and the calibrated runoff predictive models were formulated with the newly derived $\lambda$ value to represent the aforementioned decadal rainfall-runoff conditions. Inter-decadal runoff difference can then be mapped with the runoff difference between newly formulated decadal rainfall-runoff models while non-parametric statistics were used for runoff trend analyses.

This study also re-assessed the validity of Equation (1) on the entire M70, M80, and M90 datasets through the reverse derivation of the $\lambda$ value. Equation (2) will be discarded if the inferential statistics of the *P-Q* dataset reject the validity of $\lambda = 0.2$. Equation (1) will then be calibrated according to the *P-Q* data pairs of M70, M80, and M90. Non-parametric inferential statistics, bootstrapping, and Bias Corrected and Accelerated (BCa) procedure (2000 samples with replacement) were conducted in SPSS on each batch of derived $\lambda$ and $S$ values in order to search for optimum values within the confidence intervals at alpha 0.01 level. The $\lambda$ confidence intervals' span can be used to assess the null hypothesis $H_0$ as shown below:

Null Hypothesis ($H_0$): Equation (2) ($\lambda = 0.2$) is valid to model runoff estimates with DID HP 27 dataset.

The SCS-CN model calibration of this study consists of the following steps:

1. Rearrange Equation (1) into: $S = \frac{(P-I_a)^2}{Q} + I_a - P$

2. For each *P-Q* event pair, calculate the corresponding *S* value with the above equation under the SCS constraint that $I_a < P$ value and calculate the $\lambda$ value with $\lambda = I_a/S$.

3. Conduct bootstrap, BCa (at $\alpha = 0.01$ level) inferential statistical analyses (2000 samples with replacement) for the calculated $\lambda$ and *S* datasets separately for each decadal model.

4. Generate 99% confidence intervals for $\lambda$ and *S* datasets for each decadal model.

5. Test the null hypothesis ($H_0$) by referring to the $\lambda$ confidence intervals' span and its standard deviation for each decadal model. If the $\lambda = 0.2$ value exists within the $\lambda$ confidence interval, use Equation (2) to model rainfall-runoff. Otherwise, move to step 6.

6. Find the optimum $\lambda$ and *S* values from BCa confidence intervals and calculate $I_a$ for each decadal model using supervised optimization technique by minimizing the overall model bias (BIAS) near to the value of zero.

7. Formulate the calibrated SCS model by substituting $I_a$ and *S* into Equation (1).

8. According to a group of researchers [63], when $\lambda$ value other than 0.2 is detected, its corresponding *S* value (denoted by $S_\lambda$) must be correlated to the $S_{0.2}$ values for CN calculation. As such, correlate $S_\lambda$ and $S_{0.2}$ with the *S* general formula which was derived by a past researcher [50]: $S_\lambda = \frac{\left[P - \frac{(\lambda-1)Q}{2\lambda}\right] - \sqrt{PQ - P^2 + \left[P - \frac{(\lambda-1)Q}{2\lambda}\right]^2}}{\lambda}$

9. Substitute optimum $\lambda$ and $S_\lambda$ into Equation (1) to formulate the decadal model.

10. Lastly, substitute $S_{0.2} = \frac{25,400}{CN_{0.2}} - 254$ into each decadal model to express *Q* in term of *P* and $CN_{0.2}$.

Note: Appendix B shows step 9 and 10 using an example.

### 2.3. IMERG Satellite Rainfall Trend Analysis

The Integrated Multi-Satellite Retrievals for Global Precipitation Measurement (IMERG) [64] is a unified satellite product that has been launched by NASA and JAXA. This product combines, calibrates, and integrates satellite microwave precipitation estimates with microwave-calibrated infrared satellite estimates, precipitation gauge analyses, and other precipitation estimators, in order to provide precipitation measurements at fine time and space scales across the globe. There are three types of IMERG systems: Early (IMERG-E), Late (IMERG-L), and Final (IMERG-F) runs, with latency times of 6 h, 18 h, and 3 months, respectively. All IMERG runs are available at half-hourly, daily, and monthly temporal resolutions, and have global coverage at a spatial resolution of 0.1°.

In this study, the IMERG-F monthly version 6 product at a spatial resolution of 0.1° was employed, as it has been bias-corrected using precipitation gauges from the Global Precipitation Climatology Centre (GPCC) and is considered more accurate for scientific research compared with other IMERG products. The data can be accessed through NASA's Giovanni system [65]. Monthly rainfall data from 2001 to 2020 (Appendix A) was obtained from the Giovanni system to demonstrate the annual rainfall distribution across Malaysia.

Trend analysis was conducted on the monthly rainfall data in Malaysia. Before conducting the analysis, a normality test was performed to determine which central of tendency (mean or median) was to be used. If a dataset has less than 2000 samples, the Shapiro-Wilk test is recommended instead of the Kolmogorov-Smirnov test. This study used a dataset with less than 2000 samples; therefore, the Shapiro-Wilk test was employed. If the *p*-value is greater than 0.05, the dataset is considered to be normally distributed, and therefore, mean is used to measure the central tendency of the dataset [50]. The non-parametric inferential statistics using the BCa bootstrapping method was conducted on 2000 random samples (with replacement) to calculate the 99% confidence intervals of the mean or median for each interval.

Rainfall time series forecasting models were created using monthly rainfall data in Malaysia from January 2001 to December 2020 (N = 240) with the Expert Modeler in SPSS. The rainfall amount was forecasted for the next 24 months, from January 2021 to December

2022 in Malaysia, to determine if there will be a significant change in rainfall pattern in the near future that could affect runoff prediction.

## 3. Results and Discussion

### 3.1. The Optimum $\lambda$ and S of Decadal Models

Tables 1–3 display the BCa 99% confidence intervals of $\lambda$ for the M70, M80, and M90 decadal datasets. The BCa 99% confidence intervals of both the mean and median for each decadal dataset do not include the value of 0.2, leading to the rejection of the null hypothesis ($H_0$) at the alpha = 0.01 level. Equation (2) was found to be statistically invalid and, thus, cannot be utilized to model runoff conditions in Peninsula Malaysia for the M70, M80, and M90 decadal datasets. The rejection of $H_0$ necessitates the search for a new, optimal value of $\lambda$ to develop a new rainfall-runoff prediction model.

**Table 1.** Inferential statistics results of derived $\lambda$ for M70 decadal dataset at $\alpha$ = 0.01.

| $\lambda$ | Statistics | Bootstrap, BCa 99% | | | |
| | | Bias | Std. Error | Confidence Intervals | |
| | | | | Lower | Upper |
| --- | --- | --- | --- | --- | --- |
| Skewness | 3.815 | | | | |
| Kurtosis | 19.768 | | | | |
| Mean | 0.098 | 0.0003 | 0.014 | 0.069 | 0.142 |
| Median | 0.065 | 0.0006 | 0.00294 | 0.049 | 0.089 |

**Table 2.** Inferential statistics results of derived $\lambda$ for M80 decadal dataset at $\alpha$ = 0.01.

| $\lambda$ | Statistics | Bootstrap, BCa 99% | | | |
| | | Bias | Std. Error | Confidence Intervals | |
| | | | | Lower | Upper |
| --- | --- | --- | --- | --- | --- |
| Skewness | 3.217 | | | | |
| Kurtosis | 13.028 | | | | |
| Mean | 0.095 | 0.0002 | 0.014 | 0.066 | 0.135 |
| Median | 0.047 | 0.0022 | 0.007 | 0.034 | 0.064 |

**Table 3.** Inferential statistics results of derived $\lambda$ for M90 decadal dataset at $\alpha$ = 0.01.

| $\lambda$ | Statistics | Bootstrap, BCa 99% | | | |
| | | Bias | Std. Error | Confidence Intervals | |
| | | | | Lower | Upper |
| --- | --- | --- | --- | --- | --- |
| Skewness | 5.393 | | | | |
| Kurtosis | 34.674 | | | | |
| Mean | 0.076 | 0.00005 | 0.013 | 0.051 | 0.115 |
| Median | 0.042 | 0.00183 | 0.007 | 0.031 | 0.063 |

The $\lambda$ dataset is skewed and tested to be non-normally distributed in SPSS for all three decadal groups and therefore, the search for an optimal representative $\lambda$ value using a supervised optimization technique will be concentrated on the range of median confidence intervals. These intervals are [0.049, 0.089] for the M70 dataset (Table 1), [0.034, 0.064] for the M80 dataset (Table 2), and [0.031, 0.063] for the M90 dataset (Table 3).

The BCa 99% confidence intervals of $S_\lambda$ for the M70, M80, and M90 decadal datasets are presented in Tables 4–6. The normality of the $S_\lambda$ dataset was tested using SPSS for all three decadal groups, and found to be normally distributed. Therefore, the optimal $S_\lambda$ value was searched for within the range of the mean confidence intervals. There intervals are [117.083, 187.008] for M70 dataset (Table 4), [141.892, 231.088] for M80 dataset (Table 5), and [131.989, 192.939] for M90 dataset (Table 6).

**Table 4.** Inferential statistics results of derived $S_\lambda$ for M70 decadal dataset at $\alpha = 0.01$.

| $S_\lambda$ | Statistics | Bootstrap, BCa 99% | | | |
| | | Bias | Std. Error | Confidence Intervals | |
| | | | | Lower | Upper |
| Skewness | 1.298 | | | | |
| Kurtosis | 0.975 | | | | |
| Mean | 151.592 | −0.482 | 14.954 | 117.083 | 187.008 |
| Median | 123.615 | −0.166 | 11.367 | 91.255 | 157.815 |

**Table 5.** Inferential statistics results of derived $S_\lambda$ for M80 decadal dataset at $\alpha = 0.01$.

| $S_\lambda$ | Statistics | Bootstrap, BCa 99% | | | |
| | | Bias | Std. Error | Confidence Intervals | |
| | | | | Lower | Upper |
| Skewness | 1.794 | | | | |
| Kurtosis | 5.201 | | | | |
| Mean | 180.994 | −0.339 | 14.954 | 141.892 | 231.088 |
| Median | 147.950 | −3.921 | 19.112 | 113.890 | 183.670 |

**Table 6.** Inferential statistics results of derived $S_\lambda$ for M90 decadal dataset at $\alpha = 0.01$.

| $S_\lambda$ | Statistics | Bootstrap, BCa 99% | | | |
| | | Bias | Std. Error | Confidence Intervals | |
| | | | | Lower | Upper |
| Skewness | 1.132 | | | | |
| Kurtosis | 1.407 | | | | |
| Mean | 161.827 | 0.221 | 11.934 | 131.989 | 192.939 |
| Median | 142.610 | −2.566 | 21.298 | 95.236 | 189.506 |

The optimal $\lambda$ and $S_\lambda$ values for the M70, M80, and M90 decadal datasets using a supervised optimization technique are presented in Table 7. The product of the optimal $\lambda$ and $S_\lambda$ values gives the representative initial abstraction value for each dataset, which can be calculated as $I_a = \lambda S_\lambda$.

**Table 7.** Optimal $\lambda$, $S_\lambda$ and $I_a$ for M70, M80 and M90 decadal datasets.

| Dataset | Optimal $\lambda$ | Optimal $S_\lambda$ | $I_a = \lambda S_\lambda$ |
| --- | --- | --- | --- |
| M70 | 0.049 | 160 mm | 7.904 mm |
| M80 | 0.034 | 190 mm | 6.431 mm |
| M90 | 0.031 | 160 mm | 4.956 mm |

*3.2. The Decadal Rainfall-Runoff Models*

The decadal rainfall-runoff models for Peninsula Malaysia are presented in Table 8 by substituting the respective $I_a$ and $S_\lambda$ values from Table 7 into Equation (1). Equations (3)–(5) were then formulated to model the decadal rainfall-runoff conditions in Peninsula Malaysia. To further analyse decadal runoff trend across multiple rainfall depths and $CN_{0.2}$ scenarios in Peninsula Malaysia, Equations (3)–(5) need to be re-expressed in terms of $CN_{0.2}$ to benefit SCS practitioners as they are more familiar with the use of curve number [50].

**Table 8.** Decadal rainfall-runoff models for M70, M80 and M90 decadal datasets.

| Dataset | Runoff Predictive Model | Nash-Sutcliffe Index | Equation Number |
|---|---|---|---|
| M70 | $Q = \dfrac{(P-7.904)^2}{P-152.096}$ | 0.958 | (3) |
| M80 | $Q = \dfrac{(P-6.431)^2}{P-183.569}$ | 0.910 | (4) |
| M90 | $Q = \dfrac{(P-4.956)^2}{P-155.044}$ | 0.907 | (5) |

$Q$ = runoff depth (mm), $P$ = rainfall depth (mm).

To calculate $S_\lambda$ and $S_{0.2}$ for each decadal dataset, the general $S_\lambda$ formula (step 8 in methodology Section 2.2) can be used with the optimum $\lambda$ values. Through SPSS, this study identified statistically significant power function correlation between $S_\lambda$ and $S_{0.2}$ for the M70, M80, and M90 decadal datasets, which is consistent with previous research findings [66–68]. The final equations are listed in Table 9.

**Table 9.** Correlation equations between $S_\lambda$ and $S_{0.2}$ for M70, M80 and M90 decadal datasets.

| Dataset | Correlation Equation | Adjusted R-Squared | Standard Error of Estimate | Equation Number |
|---|---|---|---|---|
| M70 | $S_{0.049} = 1.184 S_{0.2}{}^{1.081}$ | 0.939 | 0.134 | (6) |
| M80 | $S_{0.034} = 1.107 S_{0.2}{}^{1.094}$ | 0.910 | 0.201 | (7) |
| M90 | $S_{0.031} = 1.179 S_{0.2}{}^{1.069}$ | 0.907 | 0.165 | (8) |

All correlations are significant at $p < 0.001$.

Equations (6)–(8) played a crucial role in converting $S_\lambda$ to $S_{0.2}$, enabling SCS practitioners to use a rainfall-runoff model with $CN_{0.2}$, which they are more familiar with. Furthermore, by establishing a correlation between the newly derived $S_\lambda$ and $S_{0.2}$, Equations (3)–(5) were modified to be expressed in $CN_{0.2}$ terms, facilitating decadal trend analyses with $CN_{0.2}$.

Equations (3)–(5) can be expressed in $CN_{0.2}$ by substituting $S_\lambda$ in Equation (1) with Equations (6)–(8), as well as the SCS-CN formula (Step 10 in methodology Section 2.2). By doing so, the decadal runoff predictive models can be re-expressed as shown in Appendix B. The resulting alternate representations for decadal runoff predictive models in Peninsula Malaysia are presented in Table 10 in term of $CN_{0.2}$.

**Table 10.** Alternate form of decadal rainfall-runoff models for M70, M80 and M90 decadal datasets.

| Dataset | Runoff Predictive Model | Equation Number |
|---|---|---|
| M70 | $Q_{0.049} = \dfrac{\left[P - 23.077\left(\frac{100}{CN_{0.2}}-1\right)^{1.081}\right]^2}{\left[P + 447.876\left(\frac{100}{CN_{0.2}}-1\right)^{1.081}\right]}$ | (9) |
| M80 | $Q_{0.034} = \dfrac{\left[P - 15.992\left(\frac{100}{CN_{0.2}}-1\right)^{1.094}\right]^2}{\left[P + 456.589\left(\frac{100}{CN_{0.2}}-1\right)^{1.094}\right]}$ | (10) |
| M90 | $Q_{0.031} = \dfrac{\left[P - 13.618\left(\frac{100}{CN_{0.2}}-1\right)^{1.069}\right]^2}{\left[P + 425.963\left(\frac{100}{CN_{0.2}}-1\right)^{1.069}\right]}$ | (11) |

### 3.3. The Decadal Runoff Trend Analyses

The decadal runoff models (Equations (9)–(11)) enable the quantification of runoff conditions for various decades under different rainfall depths ($P$) and $CN_{0.2}$ scenarios, facilitating the analysis of runoff changes. The DID HP 27 dataset contains the lowest and highest recorded rainfall depths, ranging from 20 mm to 430 mm across $CN_{0.2}$ classes from 46 to 94. Runoff difference tables can then be calculated between any two decadal models.

This study evaluated the inter-decadal runoff differences between M70 and M80, M80 and M90, and M70 and M90 in Peninsula Malaysia. The runoff amount of the earlier

decade was subtracted from the latter to determine the inter-decadal runoff difference. For instance, the inter-decadal runoff difference between M70 (Equation (9)) and M80 (Equation (10)) was calculated by subtracting the runoff amount of M70 from that of M80. A positive inter-decadal runoff difference amount indicated a larger runoff amount in M80 compared to M70 and vice versa. Statistical analyses were performed to determine significant runoff trends between different decades. This study also correlated decadal runoff changes with deforestation and urbanization data in Peninsula Malaysia.

Non-parametric Kendall's Tau b and Spearman's Rho statistics were used to evaluate the inter-decadal runoff trend in SPSS. Both statistics showed a significant positive correlation (2-tailed) at alpha = 0.01 for all inter-decadal periods, rainfall depths, and $CN_{0.2}$ classes mentioned above. This positive correlation indicates an upward trend in inter-decadal runoff, which can be visually represented in Figures 1–3. To assess the magnitude of this upward trend in each inter-decadal scenario and $CN_{0.2}$ class (ranging from 46 to 94) according to rainfall depths from 20 mm to 430 mm, Sen slopes and its collective inferential statistics were calculated. The Sen slopes and inferential statistics of all $CN_{0.2}$ classes were then analysed collectively for each inter-decadal scenario at the alpha = 0.01 level, and the results are tabulated in Tables 11–13.

**Table 11.** Inferential statistics of Sen Slopes for inter decadal runoff difference between M80 and M90.

| Sen Slopes M80 to M90 | Statistics | Bootstrap, BCa 99% | | | |
|---|---|---|---|---|---|
| | | Bias | Std. Error | Confidence Intervals Lower | Upper |
| Skewness | −0.022 | | | | |
| Kurtosis | −1.512 | | | | |
| Mean | 0.0121 | −0.00002 | 0.00225 | 0.00701 | 0.01720 |
| Median | 0.0127 | −0.00029 | 0.00422 | 0.00439 | 0.02119 |
| Std. Deviation | 0.0087 | −0.00039 | 0.00103 | 0.00588 | 0.01032 |
| Range | 0.0247 | | | | |

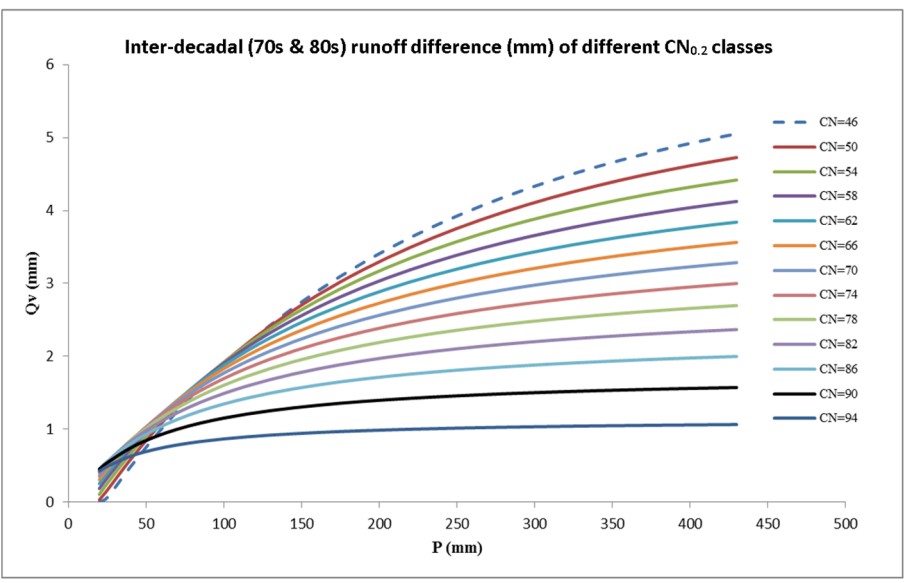

**Figure 1.** Decadal runoff difference of Equations (9) and (10) for selected $CN_{0.2}$ values to reflect the runoff change between M70 and M80 under different rainfall depths.

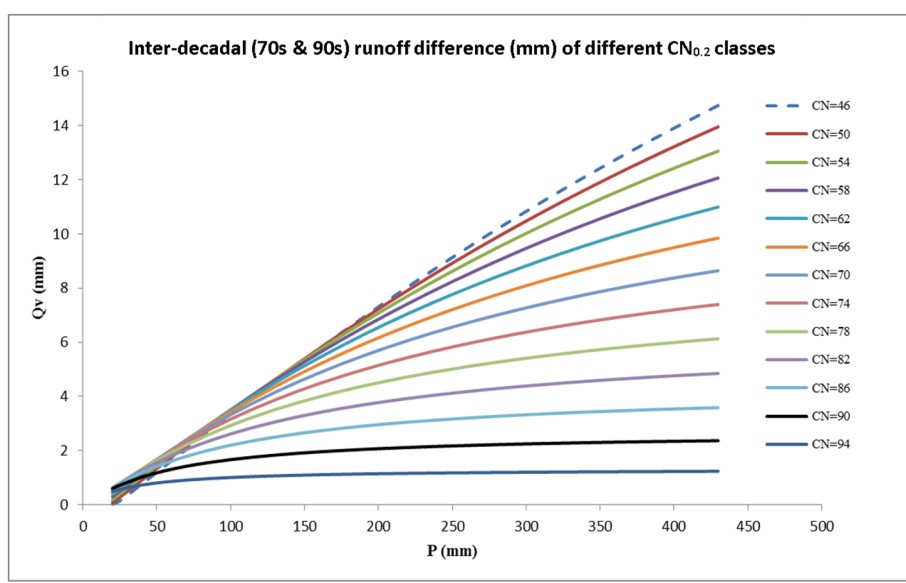

**Figure 2.** Decadal runoff difference of Equations (9) and (11) for selected $CN_{0.2}$ values to reflect the runoff change between M70 and M90 under different rainfall depths.

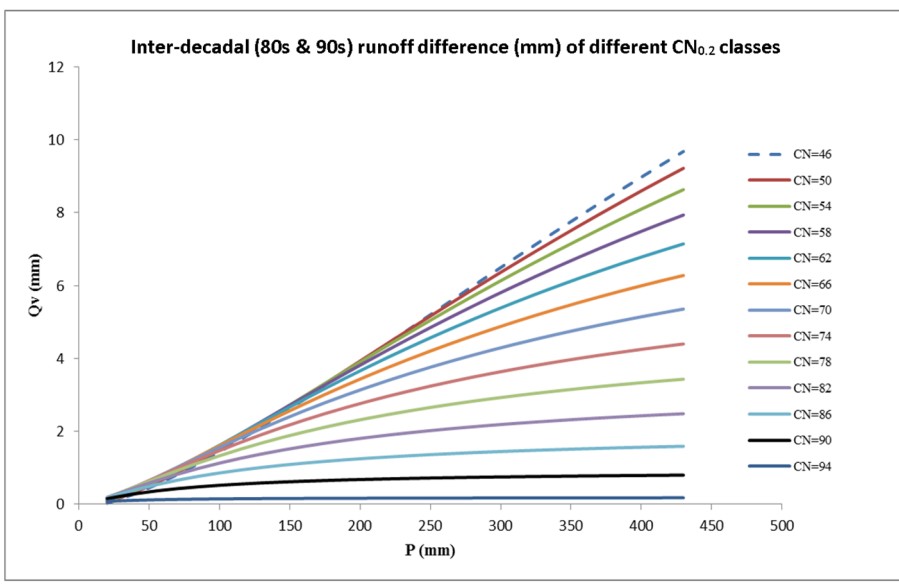

**Figure 3.** Decadal runoff difference of Equations (10) and (11) for selected $CN_{0.2}$ values to reflect the runoff change between M80 and M90 under different rainfall depths.

**Table 12.** Inferential statistics of Sen Slopes for inter decadal runoff difference between M70 and M80.

| Sen Slopes M70 to M80 | Statistics | Bootstrap, BCa 99% | | | |
|---|---|---|---|---|---|
| | | Bias | Std. Error | Confidence Intervals | |
| | | | | Lower | Upper |
| Skewness | 0.272 | | | | |
| Kurtosis | −0.940 | | | | |
| Mean | 0.0051 | −0.00001 | 0.00081 | 0.00313 | 0.00713 |
| Median | 0.0048 | 0.00008 | 0.00126 | 0.00230 | 0.00824 |
| Std. Deviation | 0.0031 | −0.00015 | 0.00046 | 0.00178 | 0.00396 |
| Range | 0.1000 | | | | |

**Table 13.** Inferential statistics of Sen Slopes for inter decadal runoff difference between M70 and M90.

| Sen Slopes M70 to M90 | Statistics | Bootstrap, BCa 99% | | | |
| | | Bias | Std. Error | Confidence Intervals | |
| | | | | Lower | Upper |
|---|---|---|---|---|---|
| Skewness | 0.065 | | | | |
| Kurtosis | −1.473 | | | | |
| Mean | 0.0178 | −0.00003 | 0.00322 | 0.01049 | 0.02506 |
| Median | 0.0175 | 0.00015 | 0.00589 | 0.00712 | 0.03125 |
| Std. Deviation | 0.0124 | −0.00057 | 0.00149 | 0.00823 | 0.01479 |
| Range | 0.0353 | | | | |

Calculated Sen slope values were tested as normally distributed in SPSS. Therefore, the mean Sen slope value was used to represent each inter-decadal runoff scenario. On average, the collective Sen slope value was 0.0121 ($p = 0.01$, 99% confidence interval from 0.00701 to 0.01720) to indicate the runoff incremental trend between M80 and M90. The Sen slope between M70 and M80 was 0.0051 ($p = 0.01$, 99% confidence interval from 0.00313 to 0.00713), while between M70 and M90 it was 0.0178 ($p = 0.01$, 99% confidence interval from 0.01049 to 0.02506). The Sen slope values also estimated the percentage of rainfall depth that becomes incremental runoff. For instance, the average expected runoff increment from a rainfall depth of 100 mm across $CN_{0.2}$ classes from 46 to 94 can be estimated to be 1.21 mm between M80 and M90 (i.e., 0.0121 × 100 mm).

The study conducted a repetition of all statistical analyses with a $CN_{0.2}$ range of 46 to 70 to assess the runoff changes across lower $CN_{0.2}$ classes, which correspond to rural and forested catchments. This was done to obtain a more accurate estimate of the inter-decadal runoff increment conditions in these areas. The results showed that the runoff incremental trend between M80 and M90 of $CN_{0.2}$ (46 to 70) had a Sen slope value of 0.0190 ($p = 0.01$, 99% confidence interval from 0.01595 to 0.02216). The Sen slope value between M70 and M80 was 0.0075 ($p = 0.01$, 99% confidence interval from 0.00606 to 0.00898), and between M70 and M90, it was 0.0276 ($p = 0.01$, 99% confidence interval from 0.02262 to 0.03239). For example, the expected runoff increment from rainfall of 100 mm across $CN_{0.2}$ classes from 46 to 70 was estimated to be 1.90 mm between M80 and M90. The study found that runoff increments were significant ($p = 0.01$) between all inter-decadal scenarios and were more apparent in forested and rural areas (highlighted area in Figure 4).

Positive inter-decadal runoff difference in Peninsula Malaysia is depicted in Figures 1–3. High rainfall depths and low $CN_{0.2}$ groups, which are associated with forested catchments, are particularly affected. These study outcomes are in line with previous studies [67–70]. Inter-decadal runoff differences are more pronounced under high rainfall depths. The mean runoff of different decades across different $CN_{0.2}$ classes was calculated and compiled, as shown in Figure 5. M90 had the highest runoff, while M70 had the lowest. Greater percentage changes in mean runoff were observed in lower $CN_{0.2}$ classes (forested catchments) compared to higher $CN_{0.2}$ classes (urban area). The largest mean runoff incremental percentage was 12.6% (6.6 mm) from M70 to M90 at $CN_{0.2} = 46$, while the smallest change was 0.1% (0.1 mm) from M80 to M90 at $CN_{0.2} = 94$.

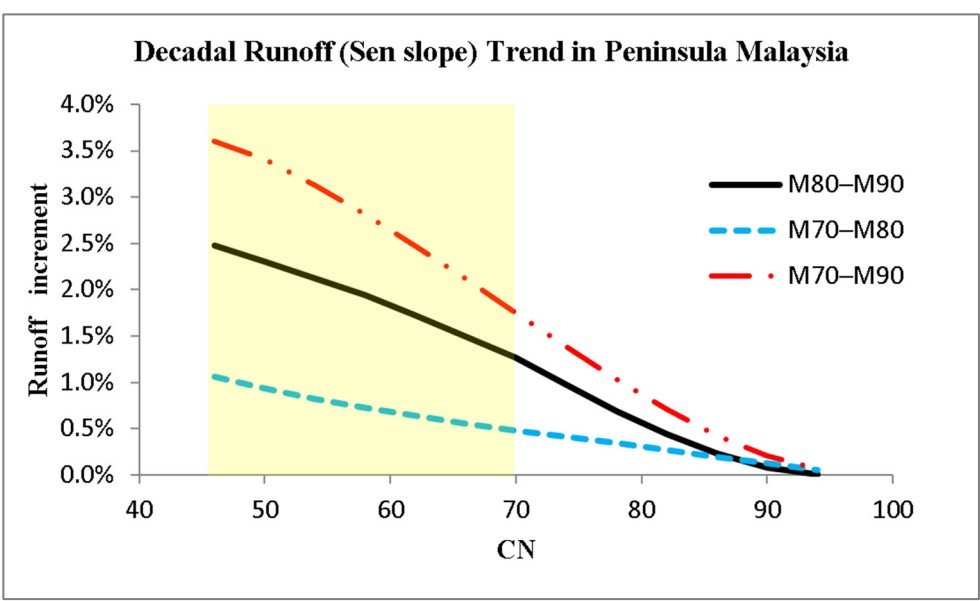

**Figure 4.** The Sen Slope for decadal runoff increment for $CN_{0.2}$ classes between M70 to M90. Sen slopes and inferential statistics were used to analyse the collective inter-decadal runoff increment conditions. Forested and rural areas are highlighted at lower $CN_{0.2}$ area from 46 to 70. The estimated percentage of rainfall depths by Sen slope calculation were compared between all scenarios to contrast the inter-decadal runoff incremental percentage.

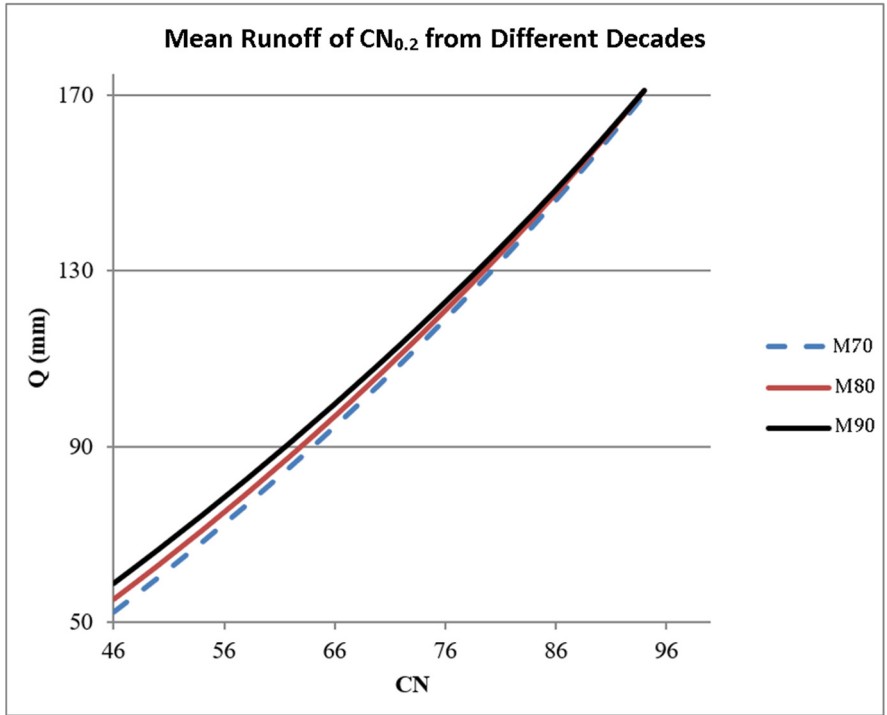

**Figure 5.** Mean runoff (mm) of decadal datasets of M70, M80, and M90 for selected $CN_{0.2}$ values.

*3.4. The Impact of $CN_{0.2}$ Variation on Runoff*

According to [71], due to variation in hydrological conditions, $CN_{0.2}$ value is often calibrated to match observed runoff dataset in modelling practice. Researchers observed that a variation of ±10% in $CN_{0.2}$ might lead to ±50% runoff variation [72] while [73] it was reported that even 1% increase in $CN_{0.2}$ with rainfall depth of 175 mm had caused 2.03% increase in runoff. References [73,74] concluded that $CN_{0.2}$ variations will have a larger impact on runoff than other parameters in Equation (1).

$CN_{0.2}$ tweaking becomes a convenient way to calibrate and validate hydrological models. However, other studies reported that $CN_{0.2}$ value of a catchment was unstable and decreased when rainfall increased [74–76]. The error and sensitivity analysis results by some researchers stated that $CN_{0.2}$ variations would induce a larger impact on runoff calculation with inherent error rather than rainfall depth variations [72,74,77]. $CN_{0.2}$ tweaking might achieve or improve temporal hydrological modelling accuracy through the trial-and-error technique, but the practicality of the end result was often uncertain and lack of statistical justification.

This study modelled the impact of $CN_{0.2}$ variation with the DID HP 27 dataset. According to [78], the practical CN values were likely to be within the range of 40 to 98. The optimum best collective $CN_{0.2}$ was 71 for the entire DID HP 27 dataset, thus, $CN_{0.2}$ variation up to 40% was chosen to cover the range of $CN_{0.2}$ from 43 to 99 and rainfall from 20 mm to 430 mm. $CN_{0.2}$ upscaling induced larger runoff change than downscaling while both effects were largely felt at rainfall depths below 100 mm. On average, runoff would reduce by 37% when $CN_{0.2}$ was downscaled up to 40% between 20 mm and 430 mm. On the other hand, the average runoff increased by 306% when $CN_{0.2}$ was upscaled up to 40%. The average runoff for both scenarios was almost identical when rainfall depths were limited to higher rainfall depths (100 mm to 430 mm). The average runoff reduced by 34% when $CN_{0.2}$ was downscaled up to 40%, while average runoff increased by 35% when $CN_{0.2}$ was upscaled to the same range. Varying the $CN_{0.2}$ value by ±10% resulted in an average runoff change of 40%, which is consistent with the findings reported in [72]. Similarly, upscaling the $CN_{0.2}$ value by 1% with a rainfall depth of 175 mm caused a 2% increase in runoff, which matches the range reported by [73]. Sen slope analyses showed that in both $CN_{0.2}$ upscaling and downscaling scenario, runoff reduction and incremental rates reduced toward the high rainfall depths but increased according to the $CN_{0.2}$ variation percentage. Lower rainfall depths (20 to 100 mm) had higher runoff variation percentages than higher rainfall depths (100 to 430 mm), as reported by previous studies [67–70]. Figures 6 and 7 present the overview of the impact of $CN_{0.2}$ variation on runoff with equations to estimate the percentage change in runoff.

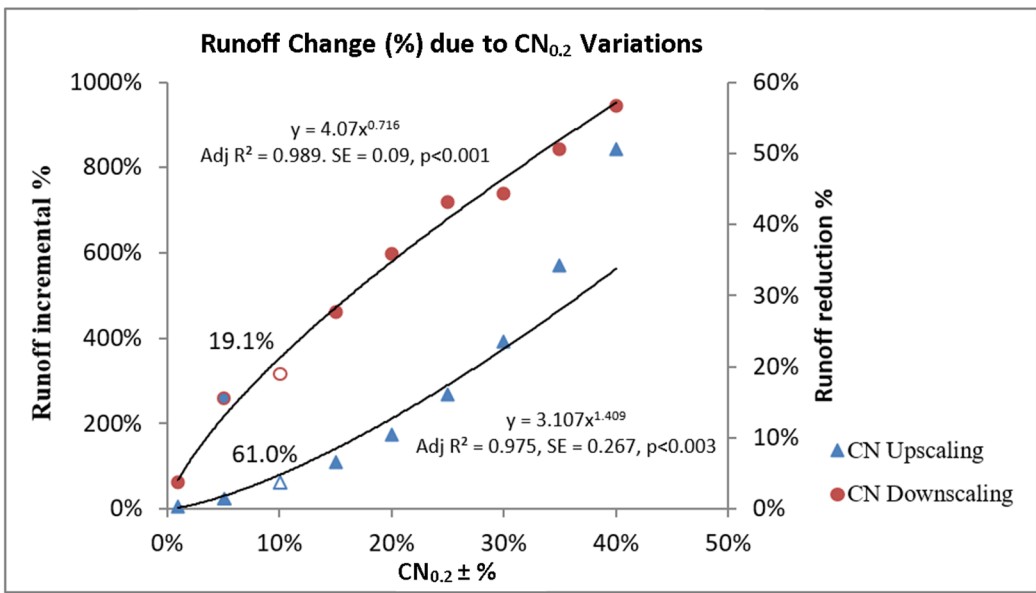

**Figure 6.** Effects of upscaling and downscaling of $CN_{0.2}$ on runoff. $CN_{0.2}$ upscaling caused runoff incremental change and vice versa. Note: $CN_{0.2}$ upscaling data points refer to primary axis. $CN_{0.2}$ variations start from $CN_{0.2}$ = 71, variation range (43 to 99) across rainfall depth range from 25 mm to 425 mm. The blank circle and triangular data point are benchmark points of $CN_{0.2}$ ± 10%, the indicated 19.1% and 61% are runoff reduction and incremental due to $CN_{0.2}$ variations.

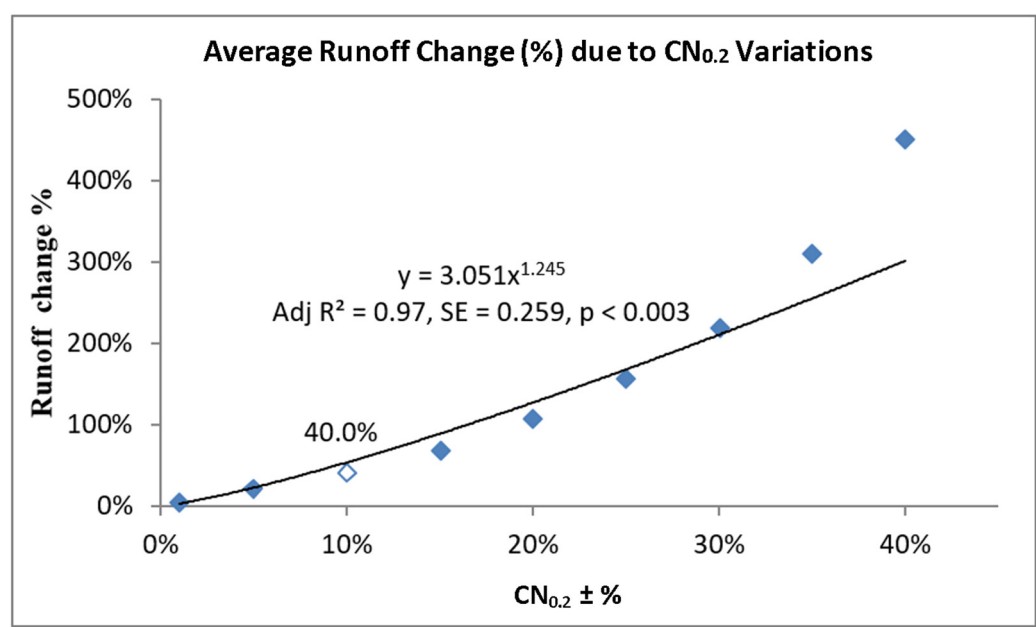

**Figure 7.** Response of runoff change to variation of $CN_{0.2}$. Runoff change % data points are the averaged runoff change due to $CN_{0.2}$ upscaling and downscaling of a specific variation %. The blank data point shows the average runoff change % due to $CN_{0.2} \pm 10\%$ variation.

### 3.5. The Impact of Deforestation and Urbanization on Runoff in Peninsula Malaysia

According to statistics from the Malaysia Department of Forestry, Peninsula Malaysia went through extensive deforestation from the 1970s. Forest area decreased at fast rates in the 1980s and started to stabilise in the 1990s. Forest area had reduced by 21.1% from M70 to M80 and 25.5% from M70 to M90 (Figure 8). Figure 9 was created to show the relationship between these decadal forest area reduction rate and its corresponding mean inter-decadal incremental runoff difference ($Q_v$%) across different $CN_{0.2}$ classes in Peninsula Malaysia. During the period between M70 and M90 in Peninsula Malaysia, the mean excess (incremental) runoff volume difference for $CN_{0.2}$ classes ranging from 46 to 70 was calculated to be 6.8 mm, equivalent to 6.8 million litres per square kilometre. This corresponds to a 10.2% increase in excess runoff, and it occurred simultaneously with a 25.5% decrease in forest area. These findings provide insights into the hydrological impacts of deforestation on non-homogeneous catchments. In general, inter-decadal mean runoff differences were more pronounced in forested and rural catchments (lower $CN_{0.2}$ classes) than urban areas. Inter-decadal runoff difference between M70 and M90 is significantly greater than runoff difference between M70 and M80 (Figure 9).

According to the published data and figures from the Department of Statistics Malaysia [79–87], the urban population in Peninsula Malaysia had been increasing rapidly (Table 14). In comparison to the forest area statistics from the Department of Forestry [53–59], an inverse correlation was identified in SPSS as:

$$FA = 5.533 + (4.922/\text{Urb-pop}) \tag{12}$$

$R^2_{adj} = 0.964$, SE = 0.175, $p < 0.012$
FA = Forest area (Million hectare)
Urb-pop = Urban population in Peninsula Malaysia (Millions)

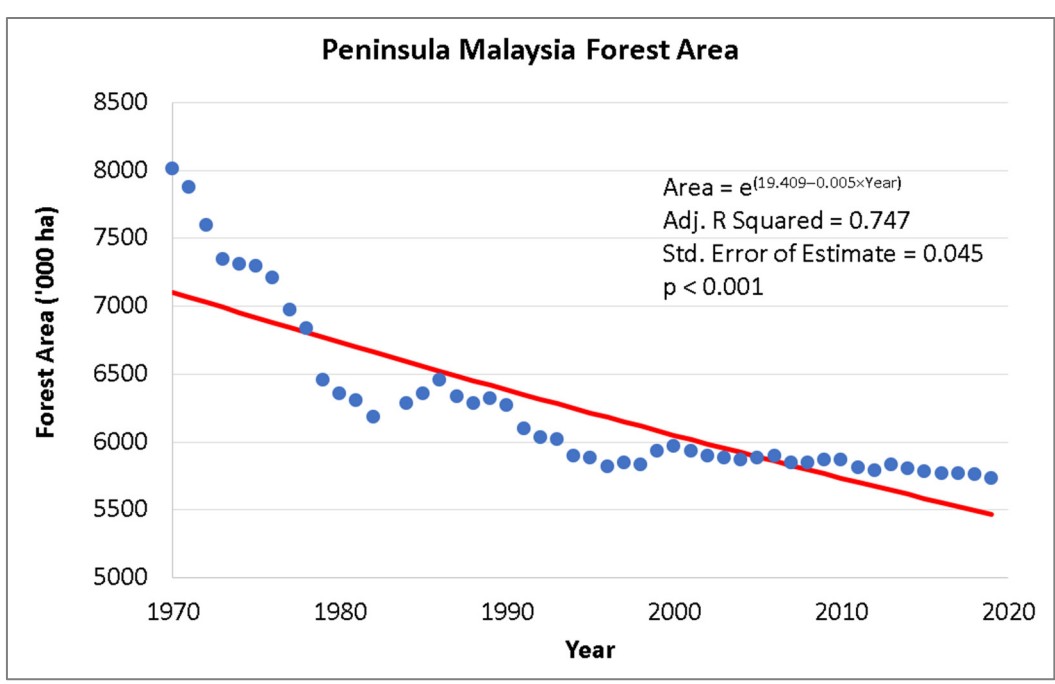

**Figure 8.** Annual statistics of forest area in Peninsula Malaysia [12,53–59,79].

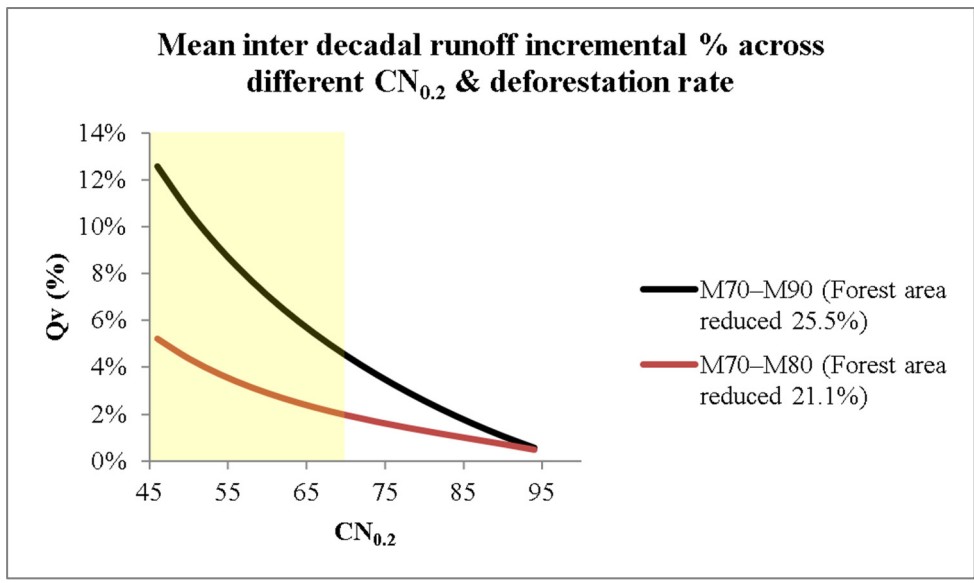

**Figure 9.** Mean inter-decadal runoff incremental % across different $CN_{0.2}$ classes (46 to 70) between 1970 (M70) and 2000 (M90). Note: The graph was created with decadal runoff models and Malaysia Department of Forestry data to coincide with the total forest area loss within the same period. On average, runoff volume for $CN_{0.2}$ classes ranging from 46 to 70 increased by 10.2% in Peninsular Malaysia while forest area reduced by 25.5% from 1970 to 2000.

The inverse correlation between urban population and the forest area in Peninsula Malaysia implies that urban development has significant correlations with deforestation (Figure 8). On the other hand, the deforestation has a direct impact on runoff amount as shown in Figure 9.

**Table 14.** Decadal urban population and forest area in Peninsula Malaysia [12,80–87].

| Year | Urban Population (Millions) | Forest Area (Millions Hectare) |
|------|---------------------------|-------------------------------|
| 1970 | 2.03 | 8.01 |
| 1980 | 4.81 | 6.35 |
| 1990 | 7.97 | 6.27 |
| 2000 | 12.26 | 5.97 |

*3.6. Decadal $\lambda$ and $I_a$*

In recent decades, urbanization in Peninsula Malaysia has caused uneven land development, leading to non-homogeneity in catchments. This study aimed to address this issue by calibrating the SCS-CN model using rainfall-runoff data from different decades to develop decadal models. The models demonstrated a strong ability to estimate runoff amounts, achieving a Nash-Sutcliffe Index ranging from 0.907 to 0.958 (Table 8), even in non-homogeneous catchments. These findings suggest that recalibrating the SCS-CN models based on regional and decadal specific rainfall-runoff conditions could be an effective approach for estimating runoff in non-homogeneous catchments.

In a previous study by the authors, the optimum $\lambda$ value for the entire DID HP 27 dataset was identified as 0.051 to model overall runoff conditions. However, in this study, different optimum $\lambda$ values were derived for the decadal datasets of M70, M80, and M90, which also led to changes in the corresponding initial abstraction ($I_a$) values (Table 7). Over time, the optimal $\lambda$ and $I_a$ values for each decade were found to decrease, indicating changes in land cover resulting from deforestation and urbanization that impact runoff conditions in rural catchments. The decreasing trend in $\lambda$ leads to a corresponding increase in runoff over time in Peninsula Malaysia.

SCS practitioners commonly calibrate $CN_{0.2}$ with one batch of runoff data and validate the final results against another batch to determine the optimum $CN_{0.2}$ value for modelling a combined dataset. However, this study highlights concerns with this practice due to land-use and land-cover changes in Peninsula Malaysia, which directly affect catchment runoff conditions over time.

There is a statistically significant upward trend in runoff (at alpha = 0.01) across all $CN_{0.2}$ classes from M70 to M90 due to changes in land use. Therefore, SCS practitioners must be cautious and aware that blindly accepting the $\lambda$ value as 0.2 is not advisable, and it is strongly recommended to derive a regional-specific $\lambda$ value. Although an optimum $\lambda$ value of 0.051 was used in a previous study [50] to model the entire dataset with a Nash-Sutcliffe value of 0.92, it differed significantly from the optimum $\lambda$ values of different decades. Hence, runoff predictive models formulated with different optimum $\lambda$ values will yield differences in runoff predictions.

*3.7. Rainfall Trend Analyses*

The BCa bootstrapping method with a 99% confidence interval was used to analyse the monthly rainfall trend in Peninsula Malaysia (Figure 10), which revealed that there has been no significant trend in the monthly rainfall over the past 20 years. The forecasted rainfall from 2021 to 2022 was consistent with the current trend. The results were supported by the model generated by Expert Modeler (Figure 11), which indicated that there would be no significant alteration in the rainfall trend in the near future.

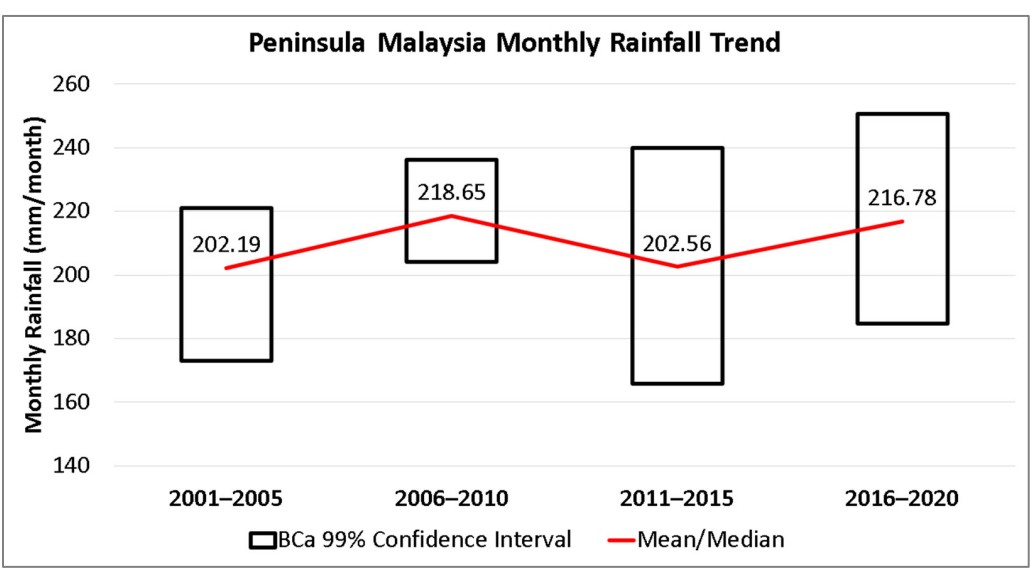

**Figure 10.** Monthly rainfall trend in Peninsula Malaysia from 2001 to 2020 (divided into 5-year interval).

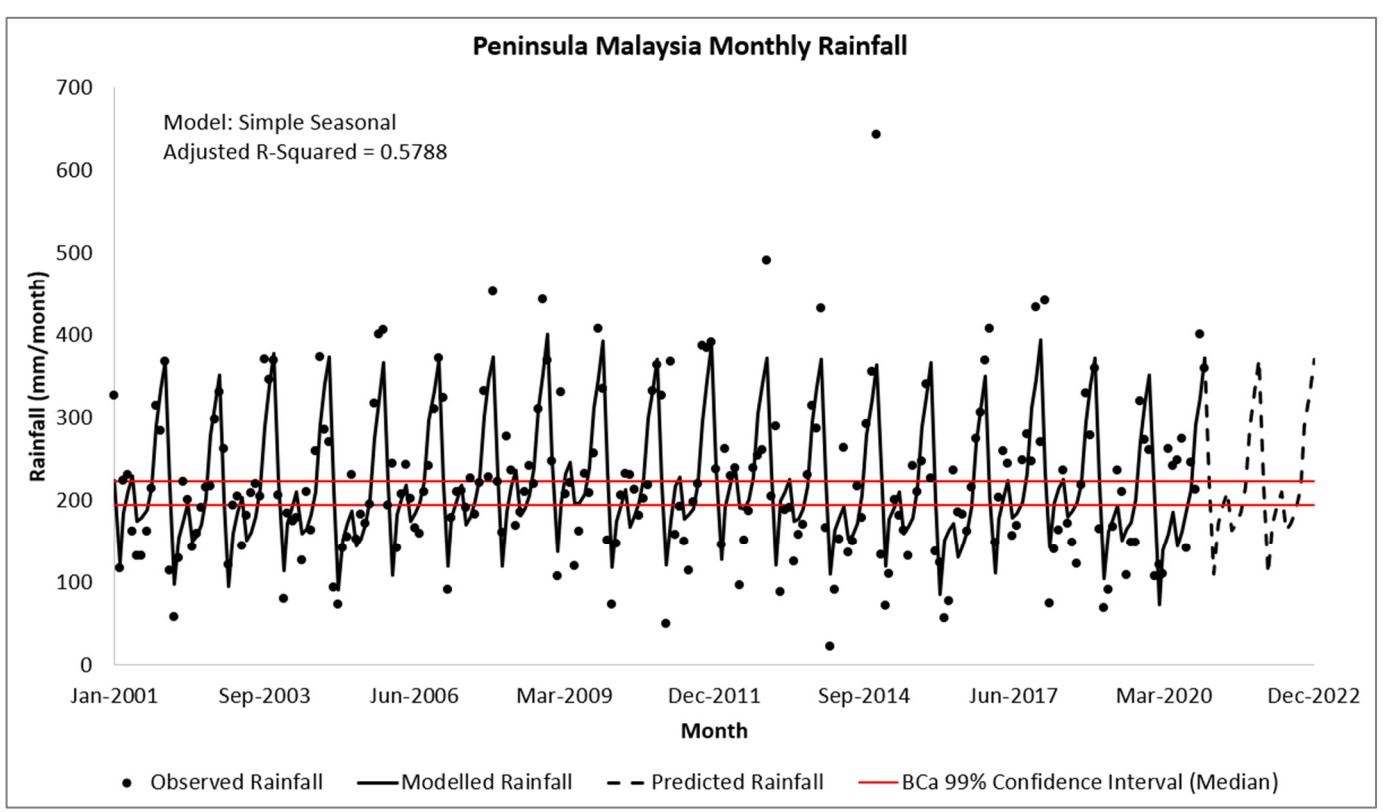

**Figure 11.** Monthly rainfall time series forecasting model for Peninsula Malaysia using Expert Modeler. Modelled period: 2001–2020 (N = 240, see Appendix A). Forecasted period: 2021–2022 (N = 24).

Flood occurrences are strongly influenced by changes in land use, including deforestation, agricultural activities, and urbanization. The conversion of natural land cover to urban and other developed land use can significantly alter the hydrological cycle, resulting in increased surface runoff and reduced infiltration. This alteration of the landscape can lead to changes in the frequency, magnitude, and timing of floods, as well as increased erosion and sedimentation in rivers and streams [37].

The increase in surface runoff resulting from the conversion of natural land for human use has caused major flooding in Malaysia [88]. Floods are the most destructive natural disasters in the country, with an estimated 85 river basins, mainly in Peninsula Malaysia, prone to recurrent flooding. Based on a study in 2014, approximately 9% of the total area of Malaysia, covering 29,800 km$^2$, is vulnerable to flood disaster, affecting almost 4.82 million people, equivalent to 22% of the total population [89]. In 2014, the states of Johor, Kelantan, Pahang, Perak, and Terengganu in Peninsula Malaysia, which were severely affected by floods, also recorded high rates of forest loss [38].

Recent floods and landslides in Malaysia have been attributed to deforestation, which results in the release of sediment and weakened soil. Trees help to prevent sediment runoffs and hold water, making them an essential factor in maintaining a stable environment. The excessive clear-cutting of trees for oil palm plantations has been identified as the primary cause of mudslides in recent times, with poor construction standards also contributing to the problem. Therefore, it can be inferred that deforestation and poor construction standards are the key factors responsible for these events, rather than El Niño or global warming, as suggested by studies [90–94].

## 4. Conclusions

This study adopted the CN hydrological model calibration technique developed by the authors in a previous study [50] and applied it in decadal runoff prediction study in Peninsula Malaysia. The correlation between forest area reduction, urbanization, and runoff volume increment was established. Highlights of the study are summarized as below:

1. The use of the conventional SCS-CN runoff model will commit type II error in this study to predict runoff conditions of different study periods. It must be pre-justified with statistics and calibrated prior to adoption for any runoff prediction. It is also not recommended to conduct calibration and validation on the entire DID HP 27 dataset of this study as each demarcated decade was represented with its unique and statistically significant runoff predictive model. Calibration and validation methodology based on the conventional SCS-CN runoff model fail to quantify accurate runoff conditions spanning across different time periods with significant land-use and cover change.

2. CN adjustment practice to formulate a hydrological model can have a large inherent error as small adjustments on the curve number can lead to large variation in the runoff. Given sufficient sample size, SCS-CN runoff model should be calibrated and formulated according to its unique optimum $\lambda$ values to represent rainfall-runoff conditions of different time periods. In this study, when CN value was varied $\pm$ 10%, the average runoff changed by 40%. This study found a significant increase in runoff across all $CN_{0.2}$ classes in Peninsula Malaysia due to changes in land use, emphasizing the importance of deriving a regional-specific $\lambda$ value and cautioning that different optimum $\lambda$ values for different decades will yield differences in runoff predictions.

3. This study emphasizes the significance of accounting for regional and decadal-specific rainfall-runoff conditions to estimate runoff in non-homogeneous catchments effectively. The calibrated SCS-CN model using data from different decades showed a remarkable ability to accurately estimate runoff amounts, even in non-homogeneous catchments. The models achieved a strong ability to estimate runoff amounts, attaining a Nash-Sutcliffe Index ranging from 0.907 to 0.958, even in non-homogeneous catchments.

4. Calibrated SCS decadal (lump) runoff models show significant decadal runoff uptrend which coincides with the overall deforestation rate in Peninsula Malaysia. The presented methodology may become more apparent with regional specific deforestation rate and its corresponding rainfall-runoff dataset. The reduction of forest area by 25.5% in Peninsula Malaysia between 1970 and 2000 was found to be directly proportional to an increase in excess runoff volume of 10.2%. In general, inter-decadal mean

runoff differences were more pronounced in forested and rural catchments (lower CN classes) than urban areas.

5.  NASA's Giovanni system was used to generate 20 years of annual rainfall maps while monthly rainfall data (2001 to 2020) was also extracted for trend analysis and short-term forecast. This study found no significant uptrend in the rainfall within the period, and the occurrence of flood and landslide incidents can likely be attributed to land-use changes in Peninsula Malaysia.

**Author Contributions:** Conceptualization, L.L.; methodology, J.F.K. and L.L.; software, J.F.K. and L.L.; validation, S.L. and L.L.; formal analysis, J.F.K. and L.L.; investigation, J.F.K. and L.L.; resources, J.F.K. and L.L.; data curation, J.F.K. and L.L.; writing—original draft preparation, J.F.K. and L.L.; writing—review and editing, J.F.K., S.L., V.L.L. and L.L; visualization, J.F.K. and L.L.; supervision, L.L.; project administration, L.L.; funding acquisition, L.L. All authors have read and agreed to the published version of the manuscript.

**Funding:** This research was supported by the Ministry of Higher Education (MoHE) through the Fundamental Research Grant Scheme (FRGS/1/2021/WAB07/UTAR/02/1) and partly supported by the Brunsfield Engineering Sdn. Bhd., Malaysia industrial grants (Brunsfield 8013/0002 & 8126/0001).

**Data Availability Statement:** Rainfall-runoff data is available at the appendix of DID, Hydrological Procedure No. 27. Design Flood Hydrograph Estimation for Rural Catchments in Peninsula Malaysia. JPS, DID, Kuala Lumpur. 2010. Available online: https://www.water.gov.my/jps/resources/PDF/Hydrology%20Publication/Hydrological_Procedure_No_27_(HP_27).pdf (accessed on 10 March 2023). IMERG-F monthly version 6 product at a spatial resolution of 0.1° data can be obtained from the NASA's Giovanni system (https://giovanni.gsfc.nasa.gov/giovanni (accessed on 15 October 2021)). Authors would like to thank Zhen Xiang Soo (Eugene) for sharing the IMERG-F data with us.

**Conflicts of Interest:** The authors declare no conflict of interest.

## Appendix A

**Table A1.** Monthly rainfall in Peninsula Malaysia from 2001 to 2009 [64].

| Year | Month | Rainfall (mm/Month) | Year | Month | Rainfall (mm/Month) | Year | Month | Rainfall (mm/Month) |
|---|---|---|---|---|---|---|---|---|
| 2001 | 1 | 326 | 2004 | 1 | 206 | 2007 | 1 | 323 |
| | 2 | 117 | | 2 | 80 | | 2 | 92 |
| | 3 | 224 | | 3 | 185 | | 3 | 179 |
| | 4 | 231 | | 4 | 174 | | 4 | 210 |
| | 5 | 162 | | 5 | 178 | | 5 | 211 |
| | 6 | 132 | | 6 | 127 | | 6 | 191 |
| | 7 | 132 | | 7 | 211 | | 7 | 226 |
| | 8 | 163 | | 8 | 163 | | 8 | 183 |
| | 9 | 214 | | 9 | 259 | | 9 | 221 |
| | 10 | 314 | | 10 | 374 | | 10 | 332 |
| | 11 | 285 | | 11 | 285 | | 11 | 229 |
| | 12 | 367 | | 12 | 270 | | 12 | 452 |
| 2002 | 1 | 114 | 2005 | 1 | 94 | 2008 | 1 | 223 |
| | 2 | 58 | | 2 | 73 | | 2 | 160 |
| | 3 | 129 | | 3 | 142 | | 3 | 278 |
| | 4 | 223 | | 4 | 155 | | 4 | 236 |
| | 5 | 200 | | 5 | 230 | | 5 | 169 |
| | 6 | 144 | | 6 | 152 | | 6 | 185 |
| | 7 | 159 | | 7 | 183 | | 7 | 210 |
| | 8 | 191 | | 8 | 172 | | 8 | 242 |
| | 9 | 216 | | 9 | 195 | | 9 | 220 |
| | 10 | 217 | | 10 | 318 | | 10 | 310 |
| | 11 | 298 | | 11 | 401 | | 11 | 443 |
| | 12 | 330 | | 12 | 406 | | 12 | 370 |

**Table A1.** *Cont.*

| Year | Month | Rainfall (mm/Month) | Year | Month | Rainfall (mm/Month) | Year | Month | Rainfall (mm/Month) |
|---|---|---|---|---|---|---|---|---|
| 2003 | 1 | 263 | 2006 | 1 | 194 | 2009 | 1 | 248 |
| | 2 | 122 | | 2 | 244 | | 2 | 107 |
| | 3 | 193 | | 3 | 142 | | 3 | 330 |
| | 4 | 204 | | 4 | 208 | | 4 | 208 |
| | 5 | 14 | | 5 | 243 | | 5 | 221 |
| | 6 | 181 | | 6 | 201 | | 6 | 120 |
| | 7 | 209 | | 7 | 167 | | 7 | 162 |
| | 8 | 220 | | 8 | 160 | | 8 | 232 |
| | 9 | 205 | | 9 | 210 | | 9 | 209 |
| | 10 | 370 | | 10 | 242 | | 10 | 257 |
| | 11 | 345 | | 11 | 310 | | 11 | 407 |
| | 12 | 369 | | 12 | 372 | | 12 | 335 |

**Table A2.** Monthly rainfall in Peninsula Malaysia from 2010 to 2018 [64].

| Year | Month | Rainfall (mm/Month) | Year | Month | Rainfall (mm/Month) | Year | Month | Rainfall (mm/Month) |
|---|---|---|---|---|---|---|---|---|
| 2010 | 1 | 151 | 2013 | 1 | 204 | 2016 | 1 | 137 |
| | 2 | 73 | | 2 | 289 | | 2 | 124 |
| | 3 | 147 | | 3 | 88 | | 3 | 57 |
| | 4 | 206 | | 4 | 188 | | 4 | 77 |
| | 5 | 232 | | 5 | 191 | | 5 | 237 |
| | 6 | 231 | | 6 | 126 | | 6 | 185 |
| | 7 | 213 | | 7 | 159 | | 7 | 182 |
| | 8 | 181 | | 8 | 171 | | 8 | 162 |
| | 9 | 202 | | 9 | 230 | | 9 | 215 |
| | 10 | 218 | | 10 | 315 | | 10 | 275 |
| | 11 | 333 | | 11 | 287 | | 11 | 305 |
| | 12 | 363 | | 12 | 432 | | 12 | 370 |
| 2011 | 1 | 327 | 2014 | 1 | 166 | 2017 | 1 | 408 |
| | 2 | 50 | | 2 | 23 | | 2 | 149 |
| | 3 | 368 | | 3 | 90 | | 3 | 203 |
| | 4 | 158 | | 4 | 153 | | 4 | 259 |
| | 5 | 193 | | 5 | 264 | | 5 | 245 |
| | 6 | 151 | | 6 | 136 | | 6 | 156 |
| | 7 | 114 | | 7 | 150 | | 7 | 169 |
| | 8 | 198 | | 8 | 217 | | 8 | 248 |
| | 9 | 220 | | 9 | 179 | | 9 | 281 |
| | 10 | 388 | | 10 | 292 | | 10 | 247 |
| | 11 | 384 | | 11 | 355 | | 11 | 433 |
| | 12 | 392 | | 12 | 643 | | 12 | 271 |
| 2012 | 1 | 238 | 2015 | 1 | 134 | 2018 | 1 | 442 |
| | 2 | 146 | | 2 | 71 | | 2 | 74 |
| | 3 | 262 | | 3 | 11 | | 3 | 140 |
| | 4 | 230 | | 4 | 201 | | 4 | 164 |
| | 5 | 239 | | 5 | 181 | | 5 | 236 |
| | 6 | 97 | | 6 | 163 | | 6 | 172 |
| | 7 | 152 | | 7 | 133 | | 7 | 149 |
| | 8 | 187 | | 8 | 242 | | 8 | 123 |
| | 9 | 239 | | 9 | 210 | | 9 | 218 |
| | 10 | 254 | | 10 | 247 | | 10 | 330 |
| | 11 | 260 | | 11 | 340 | | 11 | 279 |
| | 12 | 492 | | 12 | 226 | | 12 | 360 |

**Table A3.** Monthly rainfall in Peninsula Malaysia from 2019 to 2020 [64].

| Year | Month | Rainfall (mm/Month) |
|------|-------|---------------------|
| 2019 | 1 | 164 |
|      | 2 | 68 |
|      | 3 | 92 |
|      | 4 | 167 |
|      | 5 | 236 |
|      | 6 | 210 |
|      | 7 | 108 |
|      | 8 | 148 |
|      | 9 | 149 |
|      | 10 | 320 |
|      | 11 | 274 |
|      | 12 | 261 |
| 2020 | 1 | 108 |
|      | 2 | 121 |
|      | 3 | 110 |
|      | 4 | 262 |
|      | 5 | 241 |
|      | 6 | 249 |
|      | 7 | 275 |
|      | 8 | 142 |
|      | 9 | 246 |
|      | 10 | 213 |
|      | 11 | 401 |
|      | 12 | 360 |

**Appendix B**

Using M70 dataset as a calculation example, the optimum $\lambda$ value was identified to be 0.049 (Table 7). Substituting it into Equation (1) to obtain:

$$Q_{0.049} = \frac{(P - 0.049S_{0.049})^2}{P - 0.049S_{0.049} + S_{0.049}}$$

Substituting Equation (6) into $S_{0.049}$ in above will yield:

$$Q_{0.049} = \frac{\left[P - 0.049\left(1.184S_{0.2}{}^{1.081}\right)\right]^2}{P + 0.951\left(1.184S_{0.2}{}^{1.081}\right)}$$

Substituting $S_{0.2} = (25{,}400/CN_{0.2}) - 254$ into $S_{0.2}$ in above to obtain:

$$Q_{0.049} = \frac{\left[P - 0.049\left\{1.184\left(\frac{25{,}400}{CN_{0.2}} - 254\right)^{1.081}\right\}\right]^2}{P + 0.951\left\{1.184\left(\frac{25{,}400}{CN_{0.2}} - 254\right)^{1.081}\right\}}$$

$$Q_{0.049} = \frac{\left[P - 23.077\left(\frac{100}{CN_{0.2}} - 1\right)^{1.081}\right]^2}{\left[P + 447.876\left(\frac{100}{CN_{0.2}} - 1\right)^{1.081}\right]} \tag{A1}$$

Equation (A1) is also subject to the constraint $P > 0.049S_{0.049}$

Or $P > 23.077\left(\frac{100}{CN_{0.2}} - 1\right)^{1.081}$ else $Q_{0.049} = 0$

$P$ = Rainfall depth (mm)

$CN_{0.2}$ = Conventional SCS tabulated curve number

$$Q_{0.049} = \text{Runoff depth (mm) of } \lambda = 0.049 \text{ for M70 dataset.}$$

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
