# Peer review of "Assessing the Impact of Deforestation on Decadal Runoff Estimates in Non-Homogeneous Catchments of Peninsula Malaysia"

_water, doi:10.3390/w15061162_

Round 1

Reviewer 1 Report

I have the following comments/suggestions that should be incorporated/ addressed.

1.      Title: It is not clear, "The anthropogenic impact on decadal runoff…" How can impact be performed on decadal runoff predictions they measured.

2.      In line 311, authors should state what the version of SPSS.

3.      Authors should use subscripts of letters/ symbols/ numbers, wherever applicable in this manuscript.

4.      How has the data been collected? The instrument used for data collection, and their calibration details should be discussed.

Author Response

Thank you for taking the time and effort to review our manuscript. The comments and suggestions are very much appreciated. Please find our reply from the attached PDF file. 

Reviewer 2 Report

I found your manuscript hardly comprehensible as a result of diverse deficits which I addressed in my comments, annotated directly in the PDF attached.

Author Response

Thank you for taking the time and effort to review our manuscript. The comments and suggestions are very much appreciated. Please find our reply from the attached PDF file. We revised and restructured our manuscript. Thank you for your feedback. 

Round 2

Reviewer 2 Report

You substantially improved this manuscript. I'd agree publishing the revised version.